# The Effect of Data Leakage and Feature Selection on Machine Learning Performance for Early Parkinson’s Disease Detection

**DOI:** 10.3390/bioengineering12080845

**Published:** 2025-08-06

**Authors:** Jonathan Starcke, James Spadafora, Jonathan Spadafora, Phillip Spadafora, Milan Toma

**Affiliations:** Department of Osteopathic Manipulative Medicine, College of Osteopathic Medicine, New York Institute of Technology, Old Westbury, NY 11568, USA; jstarcke@nyit.edu (J.S.); jspadafo@nyit.edu (J.S.); jspada01@nyit.edu (J.S.); pspadafo@nyit.edu (P.S.)

**Keywords:** machine learning, Parkinson’s Disease, data leakage, early diagnosis, feature selection, clinical validation

## Abstract

If we do not urgently educate current and future medical professionals to critically evaluate and distinguish credible AI-assisted diagnostic tools from those whose performance is artificially inflated by data leakage or improper validation, we risk undermining clinician trust in all AI diagnostics and jeopardizing future advances in patient care. For instance, machine learning models have shown high accuracy in diagnosing Parkinson’s Disease when trained on clinical features that are themselves diagnostic, such as tremor and rigidity. This study systematically investigates the impact of data leakage and feature selection on the true clinical utility of machine learning models for early Parkinson’s Disease detection. We constructed two experimental pipelines: one excluding all overt motor symptoms to simulate a subclinical scenario and a control including these features. Nine machine learning algorithms were evaluated using a robust three-way data split and comprehensive metric analysis. Results reveal that, without overt features, all models exhibited superficially acceptable F1 scores but failed catastrophically in specificity, misclassifying most healthy controls as Parkinson’s Disease. The inclusion of overt features dramatically improved performance, confirming that high accuracy was due to data leakage rather than genuine predictive power. These findings underscore the necessity of rigorous experimental design, transparent reporting, and critical evaluation of machine learning models in clinically realistic settings. Our work highlights the risks of overestimating model utility due to data leakage and provides guidance for developing robust, clinically meaningful machine learning tools for early disease detection.

## 1. Introduction

Machine learning (ML) has rapidly emerged as a promising tool for the early detection and diagnosis of neurodegenerative diseases [1,2], including Parkinson’s Disease (PD) [3,4]. Numerous studies have reported high diagnostic accuracy for ML models trained on clinical and paraclinical data, often exceeding 90% when leveraging features such as tremor, rigidity, bradykinesia, and other overt motor symptoms [5]. These results have fueled optimism about the potential for ML to augment or even surpass traditional clinical assessment, particularly in resource-limited or high-throughput screening settings.

However, a critical methodological challenge in the development and evaluation of ML models for clinical diagnosis is the risk of data leakage, i.e., the inadvertent use of information during model training that would not be available at the time of prediction in real-world scenarios [6]. Data leakage can arise from improper data splitting, inclusion of post-diagnostic features, or subtle correlations that artificially inflate model performance [7]. When present, leakage leads to overestimation of a model’s true predictive utility, undermining both scientific validity and clinical safety [8].

In the context of PD, most published ML models derive their predictive power from features that are themselves diagnostic criteria, such as motor symptoms or scores from clinical rating scales [9,10]. While this approach can yield impressive accuracy, it does not address the more challenging and clinically relevant question: can ML models detect PD before the emergence of overt symptoms, using only subtle or prodromal indicators? This distinction is crucial, as early detection could enable timely intervention and improved patient outcomes, whereas models that simply recapitulate existing diagnostic criteria offer little added value [3,11].

The recent literature has begun to recognize this gap. For example, studies that exclude obvious motor features or focus on prodromal PD consistently report a dramatic drop in model performance, with high rates of false positives and poor specificity [12]. Moreover, many studies rely on aggregate metrics such as accuracy or F1 score, which can mask pathological model behaviors, such as defaulting to predicting all patients as diseased, that render the models clinically unusable. Confusion matrix analysis, though rarely reported, often reveals these underlying failures [13,14].

The present study was designed to systematically investigate the impact of data leakage and feature selection on the apparent and actual performance of ML models for PD diagnosis. Specifically, we constructed two experimental pipelines: one in which all overt motor symptoms and clinically obvious features were excluded to simulate a subclinical diagnostic scenario and a control in which these features were included (Figure 1). By training and evaluating nine distinct ML models (including logistic regression, LASSO, SVM, gradient boosting, XGBoost, KNN, AdaBoost, random forest, and deep neural networks) on both feature sets, we aimed to disentangle the contributions of algorithmic complexity, feature signal, and evaluation methodology to observed performance.

Our results demonstrate that, in the absence of overt features, all models exhibited superficially acceptable F1 scores but failed catastrophically in terms of specificity, misclassifying the vast majority of healthy controls as PD. This pattern was consistent across model types and persisted despite hyperparameter tuning and regularization. When overt features were reintroduced, model performance improved dramatically, confirming that the observed diagnostic failures were not due to algorithmic limitations but to the absence of strong diagnostic signals. These findings underscore the necessity of rigorous experimental design, transparent reporting of confusion matrices, and critical evaluation of model utility in clinically realistic scenarios. They also highlight the dangers of data leakage and the importance of aligning ML evaluation with real-world clinical decision-making.

This work provides a cautionary example of how data leakage and feature selection can profoundly distort the perceived utility of ML models in clinical diagnosis. By contrasting model behavior with and without access to overt diagnostic features, we reveal the limitations of current approaches and offer guidance for the development of more robust, clinically meaningful ML tools for early PD detection.

## 2. Materials and Methods

Each subsection below details a specific stage of the experimental workflow, from data preprocessing to model evaluation. All data handling steps, including feature selection and exclusion criteria, are explicitly described to prevent inadvertent data leakage and to simulate clinically realistic scenarios. The data splitting strategy, model training protocols, and hyperparameter tuning are presented with sufficient detail to enable reproduction. Evaluation metrics and visualization techniques are reported to provide an unbiased assessment of model performance. Where applicable, pseudocode and variable definitions are included to clarify implementation choices.

### 2.1. Dataset and Preprocessing

We used a structured dataset of N=2105 patients [15], each labeled as either having PD or as a healthy control. To simulate a subclinical diagnostic scenario, all features corresponding to overt motor symptoms (e.g., tremor, rigidity, bradykinesia, postural instability, speech problems, sleep disorders, constipation) and other clinically obvious indicators were intentionally excluded. Hence, the following columns were removed prior to analysis: PatientID, DoctorInCharge, UPDRS, MoCA, FunctionalAssessment, Tremor, Rigidity, Bradykinesia, PosturalInstability, SpeechProblems, SleepDisorders, Constipation. Let D denote the original dataset, and $D^{\prime }$ the dataset after column removal.

Rather than relying on automated feature selection or importance ranking algorithms to determine which features to include for early detection [16], we based these decisions on established clinical knowledge and expert consensus. This approach ensured that only features plausibly available and relevant in a true subclinical context were retained, and that the analysis reflected real-world diagnostic challenges. By prioritizing clinical input over purely data-driven selection, we aimed to avoid the inclusion of variables that, while statistically informative, would not be accessible or meaningful in early-stage patient assessment.

Categorical variables were converted to numerical format using MATLAB’s varfun(@double, X) function (MATLAB R2024a), and the resulting table was converted to a matrix for downstream modeling.

#### Variable Definitions

X: Feature matrix of size n×d (after the removal of non-predictive columns), where *n* is the number of samples and *d* is the number of retained features.y: Binary target vector of length *n*, where $y_{i}=1$ indicates PD and $y_{i}=0$ indicates healthy control.

No explicit feature normalization or standardization was applied. After preprocessing, most predictors were binary indicators or bounded clinical scores on comparable numeric ranges, and several of the primary algorithms used (e.g., tree-based models) are relatively insensitive to feature scale. We acknowledge that algorithms such as KNN and SVM can be affected by differences in feature magnitude; this is a limitation of the present study, and future work will evaluate whether incorporating scaling materially alters their performance.

To further address the possibility that latent proxies for overt motor symptoms might remain in the feature set despite explicit column removal, we conducted a manual audit of the remaining features to identify and exclude any variables that might encode downstream consequences or clinical surrogates of overt symptoms (e.g., derived scale scores, medication use, or subjective assessments indirectly reflecting motor function). Additionally, categorical and demographic variables were examined to ensure they did not serve as proxies through class imbalance or encoding patterns. While no automated causal inference method was applied to detect hidden correlations, this conservative preprocessing strategy was intended to minimize the influence of latent proxy features and simulate a clinically realistic subclinical detection scenario.

### 2.2. Data Splitting

To evaluate model generalizability, we implemented a three-way data split:Training set: 80% of the data, used for model fitting and internal cross-validation.Validation set: 10% of the data, used for hyperparameter tuning and early stopping.Test set: 10% of the data, held out for final unbiased evaluation.

The splitting was performed in two stages using stratified random sampling to preserve class balance:The initial split divided the data into training (80%) and a temporary set (20%).The temporary set was further split equally into validation (10%) and test (10%) sets.

#### Pseudocode for Data Splitting

1:Input: Feature matrix X, label vector y2:Set random seed for reproducibility3:Partition X,y into training (80%) and temp (20%) using stratified sampling4:Partition temp into validation (50% of temp) and test (50% of temp)5:Output: $X_{\mathrm{train}},y_{\mathrm{train}},X_{\mathrm{val}},y_{\mathrm{val}},X_{\mathrm{test}},y_{\mathrm{test}}$

### 2.3. Machine Learning Models

We evaluated nine machine learning models using the same data pipeline:Logistic regression;LASSO (L1-regularized logistic regression);Support vector machine (SVM);Gradient boosting;XGBoost;k-nearest neighbors (KNN);AdaBoost;Random forest;Deep neural network (DNN).

All models were implemented in MATLAB (or via MATLAB interfaces to Python libraries (version 3.12) where necessary) and trained using the training set. Hyperparameters were tuned using the validation set. For regularized models, the regularization parameter λ was selected via grid search on the validation set.

To support reproducibility, we clarify that the grid search for hyperparameter tuning included the following parameter ranges: for logistic regression and LASSO, regularization strength λ was searched over [0.001, 0.01, 0.1, 1, 10]; for tree-based models (e.g., random forest, gradient boosting, XGBoost), the maximum tree depth was varied over [3, 5, 7, 10], the number of estimators over [50, 100, 200], and learning rates over [0.01, 0.05, 0.1]. For the deep neural network, learning rates of [0.001, 0.0005, 0.0001] and hidden layer configurations such as [64], [64, 32], and [128, 64] were evaluated. All hyperparameters were tuned based on performance on the validation set.

#### Logistic Regression Mathematical Formulation

$$P\left(y=1|x\right)=\sigma \left(w^{⊤}x+b\right)$$
where $\sigma \left(z\right)=\frac{1}{1+e^{-z}}$ is the sigmoid function, w is the weight vector, and *b* is the bias term.

The model parameters (w,b) are estimated by minimizing the regularized negative log-likelihood:$$L\left(w,b\right)=-\frac{1}{n}\sum_{i=1}^{n}\left[y_{i}\log{\hat{y}}_{i}+\left(1-y_{i}\right)\log\left(1-{\hat{y}}_{i}\right)\right]+\lambda {∥w∥}_{p}^{p}$$
where p=2 for ridge (L2) and p=1 for LASSO (L1) regularization, and ${\hat{y}}_{i}=\sigma \left(w^{⊤}x_{i}+b\right)$.

### 2.4. Learning Curve Analysis

To assess the effect of training set size on model performance, we simulated learning curves by subsampling the training data at 10 evenly spaced sizes between 10 and $N_{\mathrm{train}}$ (the full training set size). For each subsample,

A model was trained on a random subset of the training data.Accuracy was computed on both the subsampled training set and the full validation set.

#### Pseudocode for Learning Curve

1:**for** each *s* in $\left\{s_{1},s_{2},\ldots ,s_{10}\right\}$
**do**2:      Randomly select *s* samples from $X_{\mathrm{train}},y_{\mathrm{train}}$3:      Train model on this subset4:      Compute training accuracy on the subset5:      Compute validation accuracy on $X_{\mathrm{val}},y_{\mathrm{val}}$6:
**end for**


### 2.5. Prediction and Evaluation Metrics

For each model and each data split (training, validation, testing), the following metrics were computed:Accuracy (Acc): Proportion of correct predictions.Precision (Prec): Proportion of positive predictions that are correct.Recall (Rec): Proportion of actual positives that are correctly identified.F1 Score (F1): Harmonic mean of precision and recall.Confusion Matrix: Counts of true positives, true negatives, false positives, and false negatives.

#### 2.5.1. Mathematical Definitions

Let TP, TN, FP, and FN denote true positives, true negatives, false positives, and false negatives, respectively.$$\begin{array}{cc} \mathrm{Accuracy} & =\frac{TP+TN}{TP+TN+FP+FN}\\ \mathrm{Precision} & =\frac{TP}{TP+FP}\\ \mathrm{Recall} & =\frac{TP}{TP+FN}\\ F1\;\mathrm{Score} & =2\cdot \frac{\mathrm{Precision}\cdot \mathrm{Recall}}{\mathrm{Precision}+\mathrm{Recall}} \end{array}$$

#### 2.5.2. Matlab-Style Pseudocode for Metric Computation

1:**function** compute_metrics($y_{\mathrm{true}},y_{\mathrm{pred}}$)2:     $TP\leftarrow \sum (y_{\mathrm{pred}}==1\wedge y_{\mathrm{true}}==1)$3:     $TN\leftarrow \sum (y_{\mathrm{pred}}==0\wedge y_{\mathrm{true}}==0)$4:     $FP\leftarrow \sum (y_{\mathrm{pred}}==1\wedge y_{\mathrm{true}}==0)$5:     $FN\leftarrow \sum (y_{\mathrm{pred}}==0\wedge y_{\mathrm{true}}==1)$6:     $\mathrm{Accuracy}\leftarrow \frac{TP+TN}{TP+TN+FP+FN}$7:     $\mathrm{Precision}\leftarrow \frac{TP}{TP+FP}$8:     $\mathrm{Recall}\leftarrow \frac{TP}{TP+FN}$9:     $F1\leftarrow 2\cdot \frac{\mathrm{Precision}\cdot \mathrm{Recall}}{\mathrm{Precision}+\mathrm{Recall}}$10:   **return**
Accuracy,Precision,Recall,F111:
**end function**


Although techniques such as the Synthetic Minority Over-sampling Technique (SMOTE) or class weighting are commonly employed to address class imbalance [17], we elected not to apply these approaches in the present study. As a clinically science-oriented research group, we keep our work grounded in clinical realities and keep clinical applicability at the forefront, ahead of purely algorithmic considerations. Our objective was to evaluate model performance under clinically realistic conditions using the raw class distributions present in the dataset. This decision was made to avoid artificially altering the decision boundary or biasing the models in ways that may not reflect actual screening settings, where the prevalence of PD is naturally low. Instead, we relied on a three-way data split with stratified sampling to preserve class proportions and used confusion matrix analysis to directly evaluate the consequences of imbalance, particularly in terms of false positive rates. This design choice allowed us to highlight the limitations of current ML approaches in early PD detection without relying on performance enhancements that may not generalize to real-world deployment.

### 2.6. Visualization and Model Diagnostics

Learning Curves: Plots of training and validation accuracy versus training set size were generated to visualize model learning dynamics and diagnose underfitting or overfitting.Metric Comparison Across Splits: For each metric (accuracy, precision, recall, F1 score), line plots compared performance across training, validation, and test sets for each model.F1 Score Focus: Dedicated plots tracked F1 scores across the three data splits, highlighting the model’s balance between precision and recall.Confusion Matrices: For each data split and model, confusion matrices were computed and visualized, providing a granular view of true/false positives and negatives and revealing systematic misclassification patterns.

#### Pseudocode for Visualization

1:Plot learning curves: training and validation accuracy vs. training set size2:**for** each metric in {Accuracy, Precision, Recall, F1} **do**3:      Plot metric across Train, Validation, Test sets4:
**end for**
5:Plot F1-score across Train, Validation, Test sets6:**for** each split in {Train, Validation, Test} **do**7:      Compute and plot confusion matrix8:
**end for**


### 2.7. Comparative Experiments with and Without Overt Features

To determine whether the observed diagnostic failures were due to algorithmic limitations or the absence of strong diagnostic signals, all models were retrained on the full dataset, including classical Parkinsonian symptoms (e.g., tremor, rigidity, bradykinesia). This served as a control to verify that the same models could achieve clinically useful performance when provided with sufficient input signal.

### 2.8. Implementation Details and Reproducibility

All analyses were performed in MATLAB R2023b. Random seeds were set using rng(42) to ensure reproducibility of data splits and model training. All code and preprocessing steps are fully documented and available upon request.

#### Variable Summary

X_mat: Numeric feature matrix after preprocessing, size n×d.y: Binary label vector, length *n*.X_train, y_train: Training set features and labels.X_val, y_val: Validation set features and labels.X_test, y_test: Test set features and labels.lr_model, lasso_model, etc.: Trained models for each algorithm.train_sizes: Vector of training set sizes used for learning curve analysis.train_acc, val_acc: Vectors of training and validation accuracy at each training size.y_train_pred_labels, y_val_pred_labels, y_test_pred_labels: Predicted labels for each data split.acc, prec, rec, f1: Computed metrics for each split.

### 2.9. Summary of Workflow

Load and preprocess data, removing overtly diagnostic columns.Convert categorical variables to numerical data and separate features from labels.Perform a stratified three-way split into training, validation, and test sets.Train nine machine learning models (logistic regression, LASSO, SVM, gradient boosting, XGBoost, KNN, AdaBoost, random forest, DNN) on the training set, tuning hyperparameters on the validation set.Simulate learning curves by subsampling the training set at multiple sizes and evaluating accuracy.Predict labels for all splits and compute accuracy, precision, recall, F1 score, and confusion matrices.Visualize learning curves, metric comparisons, F1 score trends, and confusion matrices.Repeat the above with the full feature set (including overt motor symptoms) for comparison.Save computed metrics for reproducibility and further analysis.

## 3. Results

To assess the ability of ML models to detect PD in the absence of overt clinical symptoms, we constructed two experimental pipelines. In the first, all features corresponding to classical motor symptoms (e.g., tremor, rigidity, bradykinesia) and other clinically obvious indicators were excluded, simulating a subclinical diagnostic scenario. In the second, these features were included, representing a control condition that allows for potential data leakage. Nine ML models were trained and evaluated using a three-way split (train/validation/test), and performance was assessed using accuracy, F1 score, and confusion matrix analysis.

### 3.1. Superficial Performance Masks Diagnostic Failure in the Absence of Overt Features

At first glance, several models appeared to perform reasonably well when overt features were excluded. For example, logistic regression and LASSO achieved test F1 scores of 0.74 and 0.76, respectively, while SVM and gradient boosting hovered around 0.73. However, as shown in Table 1, a closer examination revealed that all models exhibited substantial false positive rates when classifying healthy controls. Notably, LASSO and logistic regression misclassified over 90% of controls as having PD, while DNN and KNN also showed false positive rates exceeding 60%. These findings underscore a consistent diagnostic failure pattern: models achieved seemingly strong metrics by over-predicting PD in the absence of overt motor symptom features.

Although overt motor symptoms were excluded to simulate a subclinical scenario, the retained features included demographic and general health variables such as age, gender, BMI, and education level. While some of these features have been loosely associated with PD risk in prior epidemiological studies, their inclusion did not yield clinically meaningful predictive performance in our analysis. This suggests that, although subtle correlations may exist, they are insufficient in isolation to support early-stage PD classification using standard ML algorithms. These findings further emphasize the limitations of relying solely on non-diagnostic features and reinforce the clinical relevance of our negative results.

### 3.2. Overfitting and Model Collapse Across Complexity Levels

Overfitting was most pronounced in high-capacity models such as random forest, DNN, and AdaBoost, which achieved nearly perfect training scores but suffered test set F1 drops of 25–40%. Figure 2 illustrates the accuracy of all nine ML models across training, validation, and test splits. High-capacity models showed marked performance drops on unseen test data, indicative of memorization rather than learning. In contrast, simpler models like logistic regression maintained lower, but more stable, performance across splits.

Figure 3 further highlights this phenomenon by comparing F1 scores across splits. Large discrepancies between training and test F1 scores in models like DNN, random forest, and AdaBoost reveal significant generalization failures, while logistic regression and LASSO show more consistent but modest F1 scores.

### 3.3. Confusion Matrices Reveal the True Clinical Utility

Aggregate metrics such as F1 score and accuracy failed to capture the near-complete collapse of specificity exhibited by every model. As shown in Figure 4, confusion matrices for DNN and LASSO on the test set reveal that most models defaulted to predicting nearly every subject as having PD, regardless of true class. The LASSO model, for example, misclassified 118 of 120 healthy controls as having PD, despite a high F1 score. In contrast, the DNN model, while exhibiting lower F1 and accuracy scores overall, achieved the lowest false positive rate among all models. These contrasting confusion matrices emphasize the limitations of relying solely on aggregate metrics.

In addition to the high false positive rates (Type I errors), we also evaluated false negatives (Type II errors) across all models. While many models defaulted to predicting most subjects as having PD (resulting in high false positives) the impact on false negatives varied. For example, the DNN model misclassified 44 actual PD cases as healthy controls (Figure 4), representing a substantial Type II error rate despite its lower false positive rate. This pattern suggests a trade-off between overprediction and sensitivity loss, reinforcing the limitations of relying solely on metrics like the F1 score. Including overt features, as seen in the following sections, reduced both false positives and false negatives substantially, highlighting how access to diagnostically informative features artificially improves both types of error rates.

Notably, while the DNN model demonstrated lower overall F1 and accuracy scores compared to simpler models, it achieved the lowest false positive rate among all models evaluated without overt features (Figure 4). This paradoxical pattern (relatively better specificity despite poorer aggregate metrics) was also observed in models such as AdaBoost and KNN, both of which reported modest F1 scores but demonstrated improved ability to correctly identify healthy controls compared to models like LASSO or logistic regression. This phenomenon may be partially explained by differences in model architecture and regularization behavior. For instance, the DNN’s use of dropout and weight decay likely encouraged a more conservative decision boundary, reducing overconfident misclassification. Similarly, KNN’s instance-based learning approach is sensitive to local density, which may yield cautious predictions in sparsely populated control regions. AdaBoost, by emphasizing misclassified instances through iterative weighting, may have inadvertently focused on recovering specificity at the expense of recall. These findings suggest that certain model types, particularly those with non-linear representation capacity or adaptive weighting mechanisms, may be better suited to managing class imbalance in subclinical scenarios where overt signals are absent. However, this increased specificity does not necessarily equate to superior clinical utility, as most of these models still exhibited substantial false positive rates overall.

### 3.4. Model Behavior with Obvious PD Features Included: The Effect of Data Leakage

To determine whether the observed diagnostic failures were due to algorithmic limitations or the absence of strong diagnostic signals, all models were retrained on the full dataset, including classical Parkinsonian symptoms. As shown in Table 2, performance improved dramatically across all metrics. Random forest and gradient boosting achieved test accuracies exceeding 92%, with false positive rates as low as 8.3%. This sharp contrast to the >90% false positive rates observed without obvious clinical symptoms demonstrates that the same models, when given access to overtly diagnostic features, achieve artificially high performance—an effect attributable to data leakage.

Figure 5 presents confusion matrices for random forest and KNN on the test set when overt features were included. The random forest model demonstrates superior specificity, with only 29 false positives, while KNN exhibits the highest false positive count (44) among the models analyzed.

To preserve readability and avoid redundancy, we do not present confusion matrices for all nine models across the three data partitions. Visual inspection shows that the misclassification patterns are highly concordant (e.g., elevated false positive rates when overt clinical features are omitted), and these trends are already captured by the reported specificity and “percent falsely predicted PD” metrics in Table 1 and Table 2. Accordingly, we display only representative matrices that illustrate the two dominant behaviors (specificity collapse vs. more balanced errors).

### 3.5. Learning Curves Reveal the Impact of Overt Feature Inclusion

To directly illustrate the effect of including overt clinical features on model learning dynamics, we compared learning curves for LASSO logistic regression (trained without overt features) and KNN (trained with overt features). Figure 6 shows training and validation accuracy as a function of training set size for both scenarios.

In well-behaved learning scenarios, training and validation curves should gradually converge toward each other as the training set size increases, eventually plateauing at similar performance levels with only a small, stable gap between them This convergence pattern suggests that the model is learning genuine patterns rather than memorizing noise, and that additional training data helps reduce overfitting while improving generalization.

These results highlight the dramatic difference in achievable accuracy and generalization when overt clinical features are included. The LASSO model, trained without overt features, shows limited learning capacity and converges to moderate accuracy, reflecting the true difficulty of early PD detection. In contrast, the KNN model, trained with overt features, achieves much higher accuracy, but this performance is largely attributable to the inclusion of features that are themselves diagnostic, thus inflating apparent model utility.

Figure 7 presents learning curves for two representative high-capacity machine learning models (i.e., random forest (a) and AdaBoost (b)) trained on a set that explicitly excludes overtly diagnostic indicators of PD, such as tremor, rigidity, and other classical motor symptoms. In both models, the training accuracy remains at or near 1.0 (100%) across all training set sizes, indicating that the models are able to perfectly memorize the training data regardless of how much data is provided. This is a hallmark of overfitting, especially in high-capacity models with sufficient flexibility to capture even random noise in the absence of strong predictive signals.

In stark contrast, the validation accuracy for both random forest and AdaBoost plateaus at substantially lower levels, i.e., approximately 0.5 to 0.6, regardless of the amount of training data. Notably, increasing the training set size does not close the gap between training and validation accuracy; the two curves remain widely separated, with a persistent gap of 0.4–0.5. This persistent discrepancy demonstrates that the models are not learning generalizable patterns from the data, but are instead memorizing idiosyncrasies of the training set that do not translate to improved performance on unseen cases.

### 3.6. Direct Comparison and Clinical Implications

The stark contrast between model performance with and without overt features is summarized in Figure 8. This graph illustrates how excluding overt features simulates a realistic early detection scenario, resulting in low accuracy, while including overt features (data leakage) yields high but misleading accuracy. The temptation to include such features leads to artificially inflated performance metrics that do not translate to real-world clinical utility.

Hence, these results provide a cautionary example of how data leakage and feature selection can profoundly distort the perceived utility of ML models in clinical diagnosis. Only by rigorously excluding diagnostic features and critically evaluating model behavior using confusion matrices can the true limitations and potential of ML for early PD detection be revealed.

While statistical comparisons between ML algorithms (e.g., McNemar’s test or bootstrap-based confidence intervals) are common in model benchmarking studies, such analyses were intentionally omitted here. Our primary aim was not to identify statistically superior models but to evaluate whether any model could demonstrate clinically meaningful performance in the absence of overt motor features. Given that all models exhibited unacceptably high false positive rates under realistic conditions, comparative statistical testing between them would offer limited practical value. Instead, we emphasize clinical thresholds of utility (such as false positive and negative rates) over small statistical differences between algorithms, as even the “best” model under these conditions failed to reach a level of diagnostic performance that would justify clinical deployment.

## 4. Discussion

In designing and interpreting this study, our perspective was consistently guided by the realities and priorities of clinical medicine. As our research group is rooted in clinical science, our priorities differ from those of purely computational or engineering teams: rather than approaching the work solely from a computational or algorithmic vantage point, we maintained a focus on clinical translation, ensuring that our experimental decisions (such as the choice of features, validation methods, and performance metrics) were all selected to mirror the constraints and needs of real-world healthcare environments. This clinical orientation shaped our entire workflow, motivating us to move beyond technical performance and instead interrogate whether the specific ML models we evaluated in this study can genuinely contribute to earlier and more accurate diagnosis in practical, patient-facing settings.

ML holds promise for advancing medical diagnostics, but its true predictive power can only be demonstrated when models are trained on data that do not include features directly indicative of the target pathology [6,18]. If input data contain variables that are themselves diagnostic (such as overt clinical signs or test results that would immediately reveal the disease to a clinician), the resulting model may appear highly accurate, but this performance is misleading and does not reflect genuine predictive capability [19,20]. Instead, such models risk simply replicating existing diagnostic criteria rather than uncovering novel patterns or providing early detection [21,22]. As ML-based solutions become increasingly available for clinical use, it is essential for medical professionals to critically evaluate these tools and understand the nature of the data used in their development [23]. This requires learning to ask the right questions about which features were included, how the models were validated, and whether the reported performance truly reflects real-world predictive utility rather than artificial inflation due to data leakage or circular reasoning [24,25].

### 4.1. Analysis of Learning Curve Patterns

The two learning curves in Figure 6 reveal dramatically different behaviors that illuminate the impact of feature selection on model learning dynamics. The LASSO model (Figure 6a) demonstrates healthy learning curve characteristics. Both training and validation curves converge and plateau together at approximately 0.62–0.64 accuracy, with a minimal gap between them (typically less than 0.02). This convergence pattern indicates that the model has reached its true learning capacity given the available features and is not overfitting to the training data [26]. The small, stable gap between curves suggests good generalization—the model performs similarly on both seen and unseen data. However, the modest plateau level (around 62–64% accuracy) reflects the inherent difficulty of early PD detection when relying solely on legitimate, non-diagnostic features [27].

In stark contrast, the KNN model (Figure 6b) exhibits problematic learning curve behavior that violates the principles of healthy model learning [28]. The training and validation curves fail to converge, maintaining a persistent, large gap of approximately 0.15–0.17 (15–17 percentage points) even as training set size increases to 1400 samples. This behavior is characteristic of overfitting and suggests that the model is memorizing training-specific patterns rather than learning generalizable relationships [29]. The consistently high training accuracy (85–86%) combined with substantially lower validation accuracy (68–71%) indicates that the model’s apparent performance is artificially inflated by its ability to exploit the overt clinical features that directly encode the diagnostic outcome.

The failure of curves to converge in the KNN model is particularly concerning because it suggests that even with substantial increases in training data, the model cannot achieve the generalization performance that its training accuracy would suggest [30]. This persistent gap is a hallmark of data leakage, where the model has access to information that would not be available in real-world deployment scenarios [31]. The overt features essentially provide “shortcuts” to the diagnosis, allowing the model to achieve high training performance without learning the subtle, early indicators that would be clinically valuable [32].

A similar pattern is observed in high-capacity models such as random forest and AdaBoost (Figure 7), both trained without overtly diagnostic features. These models consistently achieve near-perfect performance on the training data, regardless of how much data is provided, reflecting their ability to memorize even subtle or irrelevant patterns. However, this apparent success does not translate to improved performance on new, unseen cases: their validation results remain substantially lower, and the gap between training and validation performance persists even as the training set grows. This persistent discrepancy is a classic indicator of overfitting, underscoring that, in the absence of strong predictive signals, such models are prone to capturing noise rather than learning generalizable patterns. These findings reinforce the importance of careful feature selection and robust validation [33], as high-capacity models can easily give a false impression of effectiveness if evaluated solely on their training performance.

These findings reinforce the central message of the study: high apparent accuracy in ML models for clinical diagnosis is often an artifact of data leakage or the inclusion of features that are themselves diagnostic. When such features are removed to reflect a true early detection challenge, model performance collapses, and overfitting becomes pervasive. This highlights the critical importance of proper feature selection and experimental design to avoid misleading conclusions about model utility in real-world clinical settings.

### 4.2. Strengths of the Study

This study provides a systematic investigation into the impact of feature selection and data leakage on the apparent and actual performance of ML models for PD diagnosis. By explicitly constructing parallel pipelines (with and without overtly diagnostic features) and evaluating nine distinct ML algorithms using a robust three-way data split, we offer a transparent, reproducible framework for assessing true model generalizability. The inclusion of learning curve analysis, confusion matrix visualization, and detailed metric reporting goes beyond conventional aggregate metrics, revealing subtle but clinically critical model behaviors that would otherwise remain hidden. This approach aligns with recent recommendations for standardized validation and transparent reporting in clinical AI research, and serves as a practical template for future studies seeking to avoid common methodological pitfalls [34,35].

### 4.3. Limitations and Weaknesses

As always, several limitations must be acknowledged. First, while the exclusion of overt clinical features simulates a subclinical diagnostic scenario, the remaining feature set may still contain subtle proxies for disease status, potentially introducing residual bias. Second, the dataset, though relatively large and balanced, is derived from a single source and may not capture the full heterogeneity of real-world patient populations, limiting external generalizability. Although the original dataset used in this study was relatively balanced between PD cases and controls, we recognize that real-world clinical datasets are often imbalanced. In such cases, resampling techniques (e.g., SMOTE) or class weighting may be necessary to mitigate bias in model evaluation metrics, particularly regarding false positive and false negative rates. In this study, the high false positive rates observed in the early detection scenario were due primarily to the lack of predictive signal in the available features, rather than class imbalance. Future work should systematically evaluate the impact of resampling and cost-sensitive methods, especially for deployment in settings with unbalanced prevalence. Third, this study does not include external validation on independent datasets from other institutions or geographic regions, a step increasingly recognized as essential for establishing clinical robustness [34,35]. Additionally, while multiple ML algorithms were compared, this study did not exhaustively explore all possible model architectures or ensemble strategies, nor did it incorporate advanced techniques for model interpretability or explainability. Finally, the weighting of performance metrics and the clinical relevance of specific thresholds were not formally established through expert consensus, which could further refine the evaluation of clinical utility [34].

However, it is important to note that additional regularization techniques, such as more aggressive dropout or early stopping, would not have materially altered these findings. The persistent gap between training and validation accuracy in high-capacity models reflects the intrinsic limitations of the available feature set, rather than insufficient regularization or premature stopping. This underscores that the primary barrier to improved generalization is the lack of predictive signal in the features, not the choice of model or training protocol.

A key limitation of the present study is the use of a dataset derived from a single source. While the controlled and standardized nature of this dataset allows for consistent feature collection and preprocessing, it may limit the generalizability of findings across more heterogeneous clinical populations. Differences in demographics, comorbidities, recording equipment, and diagnostic protocols could all influence model performance when applied outside the original context. As a potential next step (if suitable external data become available) we would seek to validate our models on datasets from multiple clinical sites or community-based cohorts. Such external validation would enable a more comprehensive assessment of generalizability and may reveal population- or site-specific adjustments needed for optimal model performance. Furthermore, future work could explore domain adaptation techniques and federated learning to address inter-site variability while preserving data privacy.

While recent studies have increasingly leveraged multimodal or longitudinal data (such as sensor-derived gait metrics [36], voice recordings [37,38,39], or handwriting samples [40,41]) for prodromal PD detection, our study intentionally focused on purely tabular clinical and demographic data. This design choice was driven by the desire to evaluate whether subclinical PD could be identified using low-cost, routinely collected variables that would be accessible in typical primary care or screening settings. In contrast, multimodal approaches often require specialized equipment, extensive preprocessing pipelines, or repeated longitudinal measurements that may not be feasible in all clinical environments. By deliberately restricting our feature set, we aimed to highlight the limitations and risks of overestimating model utility in early-stage PD when only minimal, non-diagnostic data are available. We acknowledge that this approach trades off sensitivity for broader accessibility and have clarified this design rationale accordingly.

### 4.4. Ensuring Clinical Value

A key lesson from this study is that the development of clinically actionable ML models requires genuine collaboration between computer science experts and medical professionals at every stage of the process. For example, in this work, rather than relying solely on automated feature selection algorithms, we deliberately used clinical expertise to determine which features should be included for early detection. This approach ensured that the retained variables were not only statistically relevant, but also plausible and meaningful in a real-world subclinical context. When ML developers work in isolation and depend exclusively on algorithmic feature selection, there is a significant risk of producing models that are technically impressive but clinically irrelevant, either by including variables unavailable at the point of care, or by missing subtle but important clinical cues. Only by integrating domain knowledge with technical innovation can we create ML tools that address real diagnostic challenges and support actionable decision-making in healthcare. This underscores the importance of interdisciplinary teamwork to bridge the gap between algorithmic performance and true clinical utility.

### 4.5. Future Directions

Building on these findings, several avenues for future research are warranted. First, future studies should prioritize external validation using datasets from diverse clinical settings to rigorously assess model generalizability and mitigate the risk of overfitting to local data idiosyncrasies [34,35]. Second, the development and adoption of standardized, clinically oriented composite metrics (such as those that integrate sensitivity, specificity, predictive values, and misclassification penalties) would provide a more nuanced assessment of model utility in real-world scenarios [34]. Third, collaboration with clinical experts to define clinically meaningful endpoints and thresholds, as well as to guide feature selection, will help ensure that ML models address genuine unmet needs rather than merely replicating existing diagnostic criteria. Fourth, future work should explore the integration of explainable AI (XAI) techniques to enhance model transparency and foster clinician trust [42], as recommended in recent reviews [35,43]. Finally, as regulatory frameworks for clinical AI continue to evolve [44,45], adherence to emerging standards for documentation, validation, and post-deployment monitoring will be critical for translating research advances into safe and effective clinical tools [34,43].

Furthermore, recent advances in deep learning for industrial fault diagnosis offer useful lessons for the challenges of early disease detection discussed in this study. For example, recent work has shown that combining time–frequency analysis methods, such as wavelet transforms, with deep learning models that can focus on important patterns in complex sensor data can greatly improve the detection of subtle faults in machinery, even when signals are noisy or variable [46]. Other research has demonstrated that integrating information from multiple types of sensor data (such as vibration and acoustic emissions) using specialized neural network architectures can further enhance the reliability of fault classification, especially in real-world industrial environments where any single signal may be ambiguous or affected by interference [47]. These trends highlight the value of using richer data representations and combining complementary sources of information to improve model robustness. While our current approach does not yet incorporate these advanced signal processing or multimodal strategies, such methods may represent promising directions for future work, particularly as wearable sensors and other non-invasive technologies continue to expand the range of physiological data available for the early detection of conditions like PD.

## 5. Conclusions

This study highlights the critical importance of rigorous experimental design, transparent reporting, and critical evaluation of machine learning models for early disease detection. By demonstrating how data leakage and feature selection can profoundly distort perceived model utility, we underscore the need for the medical AI community to move beyond superficial performance metrics and toward robust, clinically meaningful validation. Only through such efforts can the promise of ML in advancing early diagnosis and improving patient outcomes be fully realized.

If we do not begin educating our current and future medical professionals right now on how to critically evaluate and distinguish credible AI-assisted diagnostic tools from those that simply inflate their performance metrics through practices like data leakage or improper validation, we risk a future where well-intentioned clinicians adopt unreliable AI solutions in hopes of improving patient care. When these tools inevitably fail on real-world, unseen data, the resulting disappointment could foster widespread distrust of all AI-assisted diagnostics throughout the healthcare sector. Once that trust is lost, it will be difficult and time-consuming to regain. This is why it is imperative that we act now to equip medical professionals with the knowledge and skills to ask the right questions about AI validation, learning dynamics, feature selection, data leakage, and generalizability, so they can reliably identify robust, clinically meaningful AI solutions and prevent a future defined by skepticism and missed opportunities for patient care improvement.

## Figures and Tables

**Figure 1 bioengineering-12-00845-f001:**
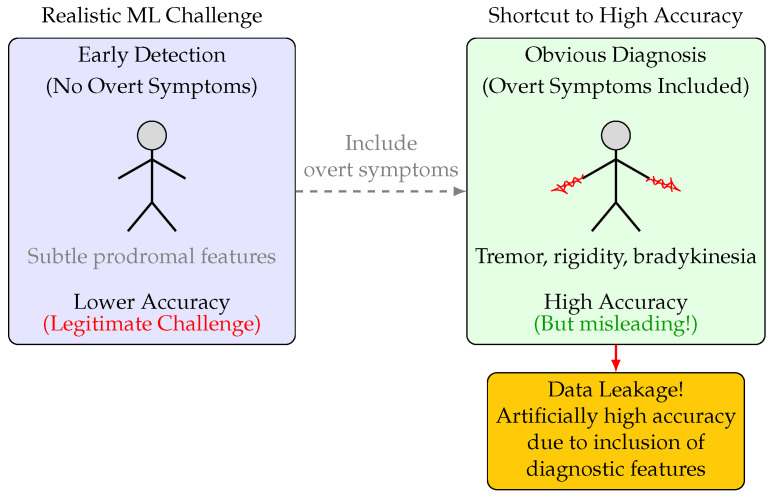
Illustration of the impact of feature selection on ML model accuracy for PD diagnosis. Left: Excluding overt motor symptoms simulates early detection, resulting in low accuracy. Right: Including overt symptoms (e.g., tremor, rigidity) yields high accuracy, but this is misleading due to data leakage—these features are themselves diagnostic and not available for true early detection. The temptation to include such features leads to artificially inflated performance metrics.

**Figure 2 bioengineering-12-00845-f002:**
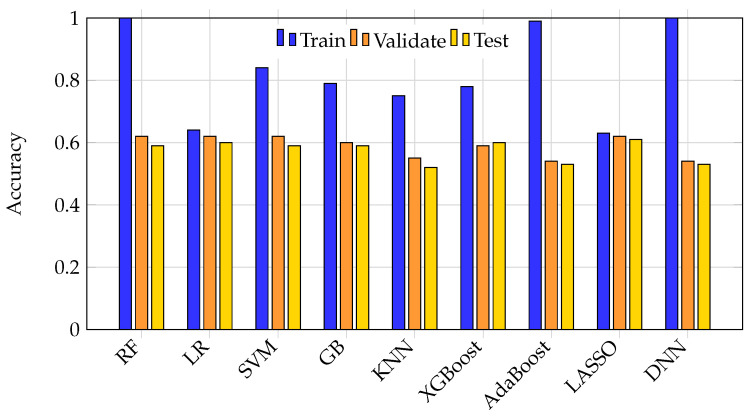
Bar graph illustrating training, validation, and test accuracy for each ML model. High-capacity models such as random forest, DNN, and AdaBoost achieve near-perfect training accuracy but show marked drops on the test set, indicating overfitting. Simpler models like logistic regression maintain lower but more consistent performance across splits.

**Figure 3 bioengineering-12-00845-f003:**
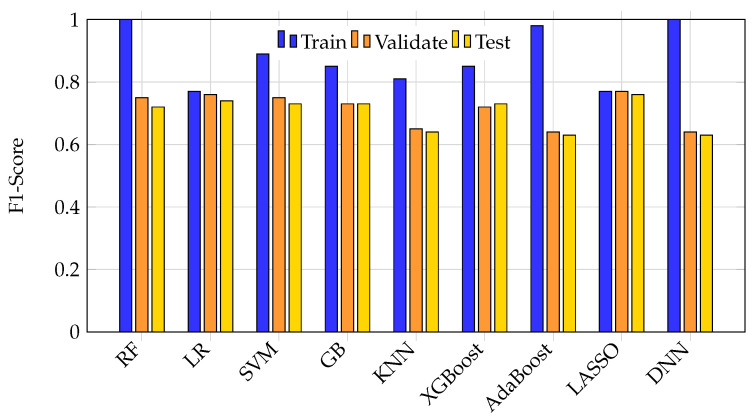
Bar graph of training, validation, and test F1 scores for each ML model. Large discrepancies between training and test F1 scores in complex models highlight generalization failure. More consistent but modest F1 scores are seen in logistic regression and LASSO.

**Figure 4 bioengineering-12-00845-f004:**
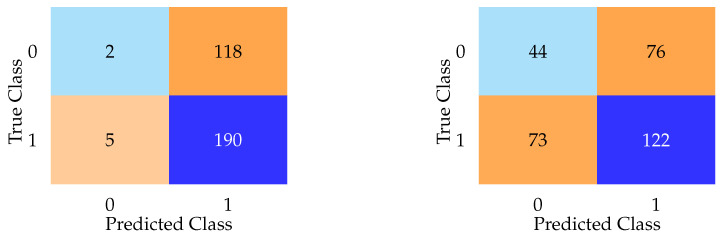
Confusion matrices highlighting diagnostic behavior of LASSO (**left**) and DNN (**right**) on the test set, with overt features excluded. The LASSO model demonstrates extreme overprediction of PD, misclassifying 111 of 120 healthy controls as having PD. The DNN model, while less accurate overall, achieves the lowest false positive rate among all models. These results illustrate how confusion matrices can reveal pathological model behavior that is hidden by summary metrics.

**Figure 5 bioengineering-12-00845-f005:**
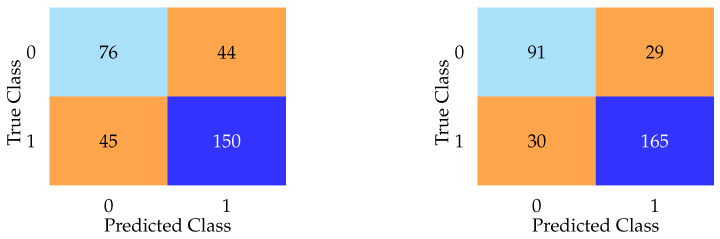
Confusion matrices for KNN (**left**) and random forest (**right**) on the test dataset when overt Parkinsonian motor features were included. The random forest model demonstrates superior specificity, with only 29 false positives, while KNN exhibits the highest false positive count (44) among the models analyzed.

**Figure 6 bioengineering-12-00845-f006:**
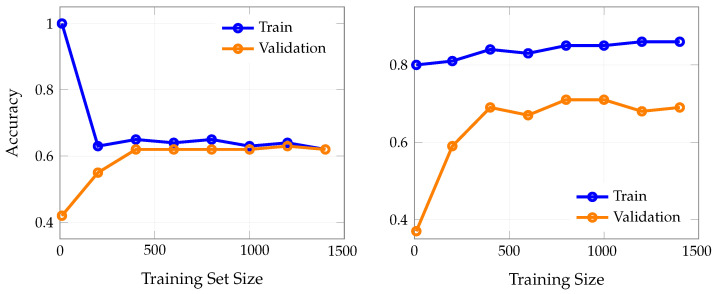
Learning curves for (**a**) LASSO logistic regression trained without overt features and (**b**) KNN trained with overt features. Excluding overt features (**a**) results in modest and plateauing accuracy, with minimal gap between training and validation curves. Including overt features (**b**) yields substantially higher accuracy, but with a persistent gap between training and validation curves, indicating potential overfitting and the artificial boost in performance due to data leakage.

**Figure 7 bioengineering-12-00845-f007:**
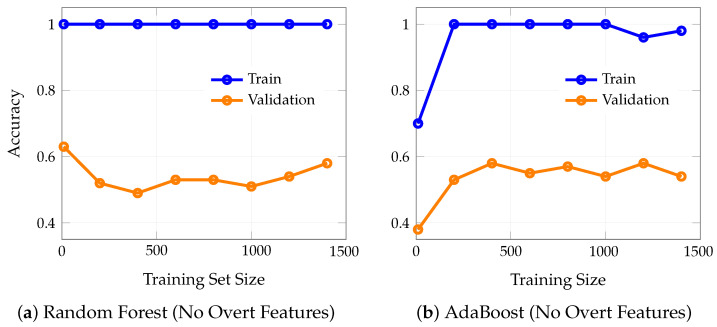
Learning curves for (**a**) random forest and (**b**) AdaBoost, both trained without overtly diagnostic features (i.e., features directly indicative of Parkinson’s Disease such as tremor or rigidity). Both models exhibit persistent overfitting: training accuracy remains near-perfect across all data sizes, while validation accuracy plateaus at substantially lower levels (approximately 0.5–0.6), with a large and stable gap between the two curves. This pattern indicates that, in the absence of strongly predictive features, high-capacity models such as random forest and AdaBoost are unable to generalize and instead memorize the training data.

**Figure 8 bioengineering-12-00845-f008:**
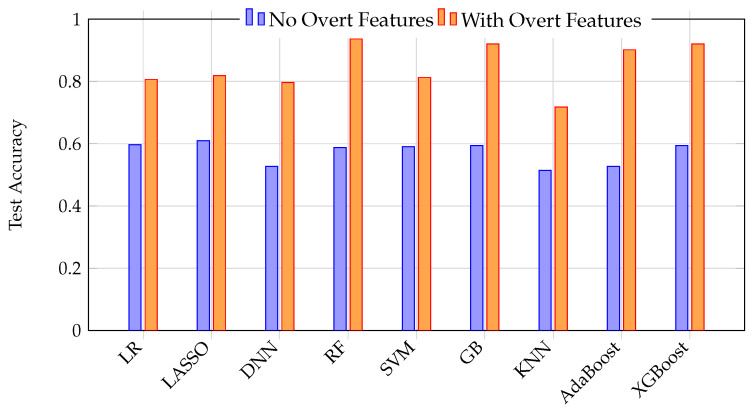
Direct comparison of test accuracy for nine ML models with and without overt features. Including overt features (orange) yields dramatically higher accuracy across all models, highlighting the risk of data leakage and the misleading nature of such results for early detection scenarios.

**Table 1 bioengineering-12-00845-t001:** Performance of ML models on the held-out test set. While F1 scores and overall accuracies varied across models, all exhibited substantial false positive rates when attempting to classify healthy controls. Notably, LASSO and logistic regression misclassified over 90% of controls as having PD, while DNN and KNN also showed false positive rates exceeding 60%. These findings underscore a consistent diagnostic failure pattern: models achieving seemingly strong metrics did so by over-predicting PD in the absence of overt motor symptom features, reinforcing the need to evaluate clinical utility beyond accuracy or F1 score.

Model	F1 Score	Accuracy	Percent Falsely Predicted PD
Logistic Regression	0.7381	0.5968	92.5%
LASSO	0.7555	0.6095	98.3%
DNN	0.6209	0.5270	63.3%
Random Forest	0.7222	0.5873	86.7%
SVM	0.7261	0.5905	87.5%
Gradient Boosting	0.7253	0.5937	85.0%
KNN	0.6450	0.5143	80.8%
AdaBoost	0.6339	0.5270	69.2%
XGBoost	0.7253	0.5937	85.0%

**Table 2 bioengineering-12-00845-t002:** Table Y: F1 scores, accuracy, and false positive rates on the test set reported for each model after retraining to include overt motor symptoms of PD. All models demonstrated substansial performance gains compared to runs excluding these features. Notably, random forest and gradient boosting achieved the highest accuracies (93.7% and 92.1%, respectively) and the lowest false positive rates (8.3% and 10.8%, respectively), indicating that clinical feature richness substantially enhances model precision and reliability.

Model	F1 Score	Accuracy	Percent Falsely Predicted PD
Logistic Regression	0.8440	0.8063	25.8%
LASSO	0.8550	0.8190	25.0%
DNN	0.8342	0.7968	25.0%
Random Forest	0.9487	0.9365	8.3%
SVM	0.8483	0.8127	24.2%
Gradient Boosting	0.9361	0.9206	10.8%
KNN	0.7712	0.7175	36.7%
AdaBoost	0.9207	0.9016	13.3%
XGBoost	0.9361	0.9206	10.8%

## Data Availability

The data presented in this study were derived from the following resources available in the public domain (accessed on 30 July 2025): https://www.kaggle.com/datasets/rabieelkharoua/parkinsons-disease-dataset-analysis [15].

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
