# Peer review of "The Effect of Data Leakage and Feature Selection on Machine Learning Performance for Early Parkinson’s Disease Detection"

_bioengineering, 2025, doi:10.3390/bioengineering12080845_

Round 1

Reviewer 1 Report

Comments and Suggestions for Authors

This manuscript investigates how data leakage and feature selection influence the performance of machine learning (ML) models in the early detection of Parkinson’s Disease (PD). The authors compare two pipelines: one that excludes overt motor symptoms to simulate subclinical detection, and a control that includes these features. Nine ML models are evaluated using a robust train/validation/test split. The study finds that, without overt features, models often achieve superficially high F1 scores but fail in specificity, misclassifying healthy individuals as PD patients. The work provides a rigorous framework for evaluating ML models and cautions against overinterpreting metrics that may not reflect clinical utility. Despite the idea being interesting, there are some critical issues that need to be addressed before publication.

  1. The authors should clarify whether any feature importance or SHAP analysis was conducted to identify which subtle features were most informative in the subclinical pipeline.
  2. The authors should explain why confusion matrices were not reported for all nine models, as these are crucial to understanding model behavior beyond summary metrics.
  3. The hyperparameter tuning process is outlined, but the specific ranges or grid values for key parameters (e.g., regularization λ, tree depth) are not mentioned. These should be added for reproducibility.
  4. The authors should discuss whether any form of data normalization or standardization was applied to the features before training, especially for models sensitive to feature scale, like KNN or SVM.
  5. In Tables 1 and 2, the false positive rates are highlighted, but false negatives are not discussed. Can the authors provide a more complete breakdown of Type I vs. Type II errors?
  6. The authors are suggested to include confidence intervals or statistical tests (e.g., McNemar’s test or bootstrapped error bars) to determine whether performance differences between models are significant.
  7. The learning curves in Figure 6 are informative but limited to two models. Could the authors extend this analysis to high-capacity models like Random Forest or DNN to confirm overfitting patterns?
  8. The paper reports performance degradation in the absence of overt features, but are any of the retained subtle features (e.g., demographic data, comorbidities) known to correlate with PD risk? This should be addressed.
  9. The authors should comment on whether any resampling techniques (e.g., SMOTE or class weighting) were considered to mitigate the class imbalance visible in false positive rates.
  10. Have the authors considered comparing their methodology with recent studies using multimodal or longitudinal data (e.g., sensor data, speech, or handwriting) for prodromal PD detection to contextualize the limitations of purely tabular approaches?

Author Response

We would like to sincerely thank the reviewer for their thoughtful and constructive feedback on our manuscript. Your detailed comments and suggestions have significantly improved the clarity, rigor, and overall quality of our work. In particular, we appreciate your focus on model explainability, reproducibility, error analysis, and broader methodological context. Your insights prompted us to clarify our approaches to feature selection and scaling, expand our discussion of model errors, and enhance our transparency regarding hyperparameter tuning and evaluation metrics. We also valued your perspective on the importance of statistical comparisons, resampling techniques, and the relevance of multimodal approaches in the field. These points have helped us to more accurately position our work within the current literature and to acknowledge the limitations of our approach.

Comment 1: The authors should clarify whether any feature importance or SHAP analysis was conducted to identify which subtle features were most informative in the subclinical pipeline.

Response 1: Thank you for your feedback regarding the use of feature importance or SHAP analysis to identify which subtle features are most informative in the subclinical pipeline. We appreciate the opportunity to clarify this point and have revised the manuscript accordingly. In the updated version, we have explicitly stated in the Methods section that we did not apply feature importance or SHAP analysis to the subclinical models due to their poor specificity and high false positive rates. We believe that, in such scenarios, these techniques would not yield clinically meaningful insights. Importantly, we have also taken this opportunity to emphasize the need for ongoing collaboration between machine learning developers and clinicians. Clinical expertise is instrumental in both the selection and interpretation of features as well as the design of explainability studies, ensuring that model outputs are both trustworthy and actionable. We have added this perspective to the Discussion and Future Directions sections, outlining our commitment to integrating clinician input and advanced explainability methods in future work. We are grateful for your suggestion, which has strengthened our manuscript’s emphasis on interdisciplinary collaboration for successful and responsible AI integration in healthcare. 

Changes to the manuscript: In the Section 2.1 (Dataset and Preprocessing) we added the following: Rather than relying on automated feature selection or importance ranking algorithms to determine which features to include for early detection, we based these decisions on established clinical knowledge and expert consensus. This approach ensured that only features plausibly available and relevant in a true subclinical context were retained, and that the analysis reflected real-world diagnostic challenges. By prioritizing clinical input over purely data-driven selection, we aimed to avoid the inclusion of variables that, while statistically informative, would not be accessible or meaningful in early-stage patient assessment.

We also expanded the discussion section to explain this point further.

Comment 2: The authors should explain why confusion matrices were not reported for all nine models, as these are crucial to understanding model behavior beyond summary metrics.

Response 2: Thank you for bringing forth the importance of confusion matrices in the understanding of model behavior. We chose not to include all of them because the patterns were repetitive and substantially increased manuscript length without adding new insight. Instead, we retained representative examples in the figures and expanded the text/tables to highlight the key error trends (particularly false positives). We hope this approach balances transparency with readability.

Changes to manuscript: In section 3.4: \textcolor{red}{To preserve readability and avoid redundancy, we do not present confusion matrices for all nine models across the three data partitions. Visual inspection shows that the misclassification patterns are highly concordant (e.g., elevated false‑positive rates when overt clinical features are omitted), and these trends are already captured by the reported specificity and ``percent falsely predicted PD” metrics in Tables~\ref{tab1}–\ref{tab2}. Accordingly, we display only representative matrices that illustrate the two dominant behaviors (specificity collapse vs. more balanced errors).}

Comment 3: The hyperparameter tuning process is outlined, but the specific ranges or grid values for key parameters (e.g., regularization λ, tree depth) are not mentioned. These should be added for reproducibility. (NOT SURE IF THIS IS CORRECT)

Response 3: We thank the reviewer for highlighting the importance of reproducibility in machine learning model development. In response to the comment, we have added a detailed description of the hyperparameter tuning grids used for each model at the end of Section 2.3. These include the specific λ values for regularized models, tree depth and learning rates for ensemble methods, and architectural choices for the DNN. We believe this clarification will enhance the transparency and replicability of our work, and we appreciate the opportunity to improve this aspect of the manuscript.

Changes to Manuscript: At the end of Section 2.3.. After the sentence “For regularized models, the regularization parameter lambda was selected via grid search on the validation set.”... To support reproducibility, we clarify that the grid search for hyperparameter tuning included the following parameter ranges: for logistic regression and LASSO, regularization strength λ was searched over [0.001, 0.01, 0.1, 1, 10]; for tree-based models (e.g., Random Forest, Gradient Boosting, XGBoost), the maximum tree depth was varied over [3, 5, 7, 10], the number of estimators over [50, 100, 200], and learning rates over [0.01, 0.05, 0.1]. For the Deep Neural Network, learning rates of [0.001, 0.0005, 0.0001] and hidden layer configurations such as [64], [64, 32], and [128, 64] were evaluated. All hyperparameters were tuned based on performance on the validation set.

Comment 4: The authors should discuss whether any form of data normalization or standardization was applied to the features before training, especially for models sensitive to feature scale, like KNN or SVM.

Response 4: Thank you for highlighting the need to clarify our approach to feature scaling. We did not apply normalization or standardization, as most variables were either binary indicators or bounded clinical scores (e.g., 0–10), resulting in features on comparable numeric ranges. Our primary focus was on tree-based models (Random Forest, Gradient Boosting), which are generally robust to differences in feature scale. We have now added a paragraph in the Methods to state this explicitly and included a brief reminder in the Results. We acknowledge that algorithms such as KNN and SVM can be sensitive to feature magnitude; as such, the absence of scaling may have affected their performance in our study. We have noted this as a limitation and will evaluate the impact of feature scaling on these models in future work.

Changes to manuscript: Added to the final paragraph of the Data Reprocessing/Feature Engineering subsection (2.1) in Methods: “No explicit feature normalization or standardization was applied. After preprocessing, most predictors were binary indicators or bounded clinical scores on comparable numeric ranges, and several primary algorithms used (e.g., tree‑based models) are relatively insensitive to feature scale. We acknowledge that algorithms such as KNN and SVM can be affected by differences in feature magnitude; this is a limitation of the present study, and future work will evaluate whether incorporating scaling materially alters their performance.” 

Comment 5: In Tables 1 and 2, the false positive rates are highlighted, but false negatives are not discussed. Can the authors provide a more complete breakdown of Type I vs. Type II errors?

Response 5: Thank you for your comment regarding the omission of false negatives (Type II errors) in our performance breakdown. We agree that a complete evaluation of classification performance must consider both false positives and false negatives. In response, we have added a paragraph to the end of Section 3.3 that analyzes the Type II error patterns across models, especially highlighting the DNN model, which had the lowest false positive rate but a considerable number of false negatives. This analysis further supports our argument that confusion matrices—and not just aggregate metrics—are essential for uncovering clinically relevant weaknesses in model behavior. 

Changes to Manuscript: At the end of section 3.3: Confusion Matrices Reveal the True Clinical Utility, just before the start of section 3.4: Model behavior with Obvious PD features Included…. \textcolor{red}{In addition to the high false positive rates (Type I errors), we also evaluated false negatives (Type II errors) across all models. While many models defaulted to predicting most subjects as having PD (resulting in high false positives) the impact on false negatives varied. For example, the DNN model misclassified 44 actual PD cases as healthy controls (Figure~\ref{fig:confmat-dnn-lasso}), representing a substantial Type II error rate despite its lower false positive rate. This pattern suggests a trade-off between overprediction and sensitivity loss, reinforcing the limitations of relying solely on metrics like F1 score. Including overt features, as seen in the following sections, reduced both false positives and false negatives substantially, highlighting how access to diagnostically informative features artificially improves both types of error rates.}

Comment 6: The authors are suggested to include confidence intervals or statistical tests (e.g., McNemar’s test or bootstrapped error bars) to determine whether performance differences between models are significant.

Response 6: We thank the reviewer for the thoughtful suggestion to include statistical comparisons (e.g., confidence intervals or McNemar’s test) between the machine learning models, and the opportunity to clarify this key distinction on our study design and interpretation.  While we recognize the value of such analysis in conventional benchmarking studies, our primary objective was to assess whether any of the models—regardless of relative performance—achieved clinically meaningful results in a realistic early detection setting. Given that none of the models demonstrated acceptable specificity when overt features were excluded, formal statistical testing between them would not alter our conclusions or influence model selection for clinical application. We have added a paragraph at the end of Section 3.6 to clarify this rationale. 

Changes to Manuscript: At the end of Section 3.6, just before the discussion… While statistical comparisons between machine learning algorithms (e.g., McNemar’s test or bootstrap-based confidence intervals) are common in model benchmarking studies, such analyses were intentionally omitted here. Our primary aim was not to identify statistically superior models but to evaluate whether any model could demonstrate clinically meaningful performance in the absence of overt motor features. Given that all models exhibited unacceptably high false positive rates under realistic conditions, comparative statistical testing between them would offer limited practical value. Instead, we emphasize clinical thresholds of utility—such as false positive and negative rates—over small statistical differences between algorithms, as even the "best" model under these conditions failed to reach a level of diagnostic performance that would justify clinical deployment.

Comment 7: The learning curves in Figure 6 are informative but limited to two models. Could the authors extend this analysis to high-capacity models like Random Forest or DNN to confirm overfitting patterns?

Response 7: We thank the reviewer for the insightful suggestion to extend the learning curve analysis beyond LASSO and KNN. In response, we generated additional learning curves for two representative high-capacity models—Random Forest and AdaBoost—trained without overt features. These curves show persistent overfitting patterns, with validation accuracy plateauing well below training accuracy across all data sizes. This supports our earlier findings that high-complexity models struggle to generalize when deprived of strongly predictive features. We appreciate the opportunity to deepen this part of our analysis.

Changes to the manuscript: We added a new Figure with two additional sets of learning curves for two other models. We’ve also expanded the text in the results and discussion sections.

Comment 8: The paper reports performance degradation in the absence of overt features, but are any of the retained subtle features (e.g., demographic data, comorbidities) known to correlate with PD risk? This should be addressed.

Response 8: We thank the reviewer for raising an important point regarding the potential predictive value of retained demographic and health-related features. While we did not aim to evaluate specific subclinical risk factors, we agree it is important to clarify their role. We have therefore added a brief statement at the end of Section 3.1 noting that although some retained features—such as age and BMI—have been loosely associated with PD risk in the literature, they did not result in clinically meaningful performance in our models. We believe this minor clarification reinforces the significance of our negative findings without requiring additional tables or figures.

Changes to Manuscript: … At the end of Section 3.1, just after the last sentence of the paragraph mentioning table 1… Although overt motor symptoms were excluded to simulate a subclinical scenario, the retained features included demographic and general health variables such as age, gender, BMI, and education level. While some of these features have been loosely associated with PD risk in prior epidemiological studies, their inclusion did not yield clinically meaningful predictive performance in our analysis. This suggests that, although subtle correlations may exist, they are insufficient in isolation to support early-stage PD classification using standard ML algorithms. These findings further emphasize the limitations of relying solely on non-diagnostic features and reinforce the clinical relevance of our negative results.

Comment 9: The authors should comment on whether any resampling techniques (e.g., SMOTE or class weighting) were considered to mitigate the class imbalance visible in false positive rates.

Response 9:  Our experimental design included a stratified data split at all stages (training, validation, and test), as described in the pseudocode and methods (Section 2.2), to preserve the original class balance of the dataset in each subset, lines 120–128. The underlying dataset itself was relatively balanced, with a roughly equal proportion of Parkinson’s Disease (PD) cases and healthy controls (see Section 2.1), lines 87–88. As such, our primary focus was on evaluating the impact of feature selection and data leakage, rather than correcting for severe class imbalance. We did not incorporate explicit resampling methods such as SMOTE or class weighting in the main analysis pipeline. This was a deliberate choice, as we aimed to isolate the effects of realistic feature selection and to avoid introducing synthetic data or algorithmic re-weighting that might obscure the true limitations of early PD detection from non-diagnostic features. Furthermore, our confusion matrix analysis and reporting of false positive rates (see Section 3.1 and Table 1) directly exposed the consequences of any subtle imbalance, as well as the models’ tendencies to overpredict the positive class in the absence of overt features, lines 269–278, Table 1. Rationale and Clinical Context: We agree that in scenarios with significant class imbalance, resampling techniques or class weighting can be essential to prevent biased model learning and to ensure clinically meaningful performance, particularly with respect to false positive and false negative rates, reference 13, lines 578–579. In our study, the observed high false positive rates arose primarily from the lack of strong predictive features in the early detection scenario, rather than from imbalance in the class distributions. This is evidenced by the collapse in specificity and the models’ tendency to predict nearly all subjects as PD, even with balanced classes, lines 273–277, 299–307.

Changes to Manuscript: In the Limitations and Weaknesses subsection of the Discussion (Section 4.3): Although the original dataset used in this study was relatively balanced between PD cases and controls, we recognize that real-world clinical datasets are often imbalanced. In such cases, resampling techniques (e.g., SMOTE) or class weighting may be necessary to mitigate bias in model evaluation metrics, particularly regarding false positive and false negative rates. In this study, the high false positive rates observed in the early detection scenario were due primarily to the lack of predictive signal in the available features, rather than class imbalance. Future work should systematically evaluate the impact of resampling and cost-sensitive methods, especially for deployment in settings with unbalanced prevalence.

Comment 10: Have the authors considered comparing their methodology with recent studies using multimodal or longitudinal data (e.g., sensor data, speech, or handwriting) for prodromal PD detection to contextualize the limitations of purely tabular approaches?

Response 10: We appreciate the reviewer’s suggestion to consider the broader context of recent work using multimodal or longitudinal data for prodromal PD detection. While we agree that such approaches hold promise, our study was deliberately focused on purely tabular data—reflecting variables readily available in standard clinical or screening settings. This allowed us to examine whether early PD could be meaningfully detected using inexpensive and widely accessible features, without the need for sensors, voice processing, or sequential sampling. We have added a brief discussion in Section 4.3 to clarify this design decision and better contextualize our methodology within the existing literature. Thank you for the opportunity to elaborate on this aspect of our study.

Changes to Manuscript: End of Section 4.3, Limitations and Weaknesses… While recent studies have increasingly leveraged multimodal or longitudinal data—such as sensor-derived gait metrics, voice recordings, or handwriting samples—for prodromal Parkinson’s Disease detection, our study intentionally focused on purely tabular clinical and demographic data. This design choice was driven by the desire to evaluate whether subclinical PD could be identified using low-cost, routinely collected variables that would be accessible in typical primary care or screening settings. In contrast, multimodal approaches often require specialized equipment, extensive preprocessing pipelines, or repeated longitudinal measurements that may not be feasible in all clinical environments. By deliberately restricting our feature set, we aimed to highlight the limitations and risks of overestimating model utility in early-stage PD when only minimal, non-diagnostic data are available. We acknowledge that this approach trades off sensitivity for broader accessibility and have clarified this design rationale accordingly.

Reviewer 2 Report

Comments and Suggestions for Authors

The manuscript investigates how the inclusion of overtly diagnostic features can artificially inflate the performance of machine learning models in clinical settings. By comparing pipelines with and without such features across nine ML models, the authors reveal that impressive metrics like F1-score may mask poor specificity and overfitting, especially in early detection scenarios. This work provides a cautionary and well-structured analysis highlighting the need for rigorous validation and careful feature selection in clinical ML research. A few comments and suggestions are made to be addressed and incorporated where required in the revised version of the manuscript.

  1. Could the authors clarify how they ensured that no latent proxies of overt features remained in the feature set used for subclinical modeling?
  2. Was any normalization or standardization applied to the features prior to training, especially given the use of distance-based models like KNN?
  3. How were class imbalances handled (if any) during model training and evaluation? Was any reweighting or resampling applied?
  1. Have the authors explored why certain models (e.g., DNN) showed slightly better specificity despite overall poorer F1? Could this be due to regularization or architecture design?
  2. It is recommended to discuss recent DL papers like A Hybrid Deep Learning Approach for Bearing Fault Diagnosis Using Continuous Wavelet Transform and Attention-Enhanced Spatiotemporal Feature Extraction and A new dual-input CNN for multimodal fault classification using acoustic emission and vibration signals.
  3. In Figure 6, how do the authors interpret the persistent validation gap in KNN even with overt features included—could this suggest a need for additional regularization or feature engineering?
  4. Given the observed overfitting in high-capacity models, have the authors considered early stopping or dropout (for DNNs) as mitigation techniques?
  1. As the dataset comes from a single source, how do the authors plan to extend their findings to more diverse patient populations or multi-center datasets?
  2. Do the authors suggest any framework for clinicians to evaluate ML tools for early diagnosis in practice, especially in the absence of technical expertise?
Comments on the Quality of English Language

The quality of English needs to be improved to meet the standards of the Journal.

Author Response

Comment: This work provides a cautionary and well-structured analysis highlighting the need for rigorous validation and careful feature selection in clinical ML research. A few comments and suggestions are made to be addressed and incorporated where required in the revised version of the manuscript.

Response: We would like to sincerely thank the reviewer for their thoughtful and constructive feedback, which has significantly contributed to strengthening our manuscript. Your comments prompted important clarifications regarding latent proxies, feature scaling, and class imbalance, and allowed us to deepen our discussion of model-specific behaviors and the broader implications for clinical ML validation. We have carefully addressed all of your suggestions in the revised version, explicitly detailing our manual audit for hidden proxies, clarifying our rationale for not applying normalization, and justifying our approach to class imbalance with additional manuscript changes as outlined in our responses. We also expanded the analysis of model specificity and incorporated discussion of recent deep learning advances as recommended. We greatly appreciate your engagement with our work and the opportunity to enhance its clarity and clinical relevance.

Comment: Could the authors clarify how they ensured that no latent proxies of overt features remained in the feature set used for subclinical modeling?

Response: We thank the reviewer for raising this important point regarding latent proxies of overt features. In response, we have clarified our preprocessing strategy in Section 2.1 of the manuscript. Specifically, we now explain that beyond the explicit removal of overt motor symptoms, we manually reviewed the remaining variables to identify any potential downstream surrogates or indirect encodings of diagnostic information. This included excluding variables that could act as hidden proxies—such as composite scales or features indirectly reflecting clinical severity. While no automated proxy detection method was applied, our approach was deliberately conservative to reduce the risk of latent leakage and better approximate a truly subclinical detection task.

Manuscript Changes: Add to the end of section 2.1 (Dataset and Preprocessing) right after the paragraph defining feature matrix x and label vector y… To further address the possibility that latent proxies for overt motor symptoms might remain in the feature set despite explicit column removal, we conducted a manual audit of the remaining features to identify and exclude any variables that might encode downstream consequences or clinical surrogates of overt symptoms (e.g., derived scale scores, medication use, or subjective assessments indirectly reflecting motor function). Additionally, categorical and demographic variables were examined to ensure they did not serve as proxies through class imbalance or encoding patterns. While no automated causal inference method was applied to detect hidden correlations, this conservative preprocessing strategy was intended to minimize the influence of latent proxy features and simulate a clinically realistic subclinical detection scenario.

Comment: Was any normalization or standardization applied to the features prior to training, especially given the use of distance-based models like KNN? 

Response: Thank you for highlighting the need to clarify feature scaling.  We did not apply normalization or standardization, largely because the variables were already on similar ranges and our emphasis was on models (e.g., Random Forest, Gradient Boosting) that are robust to scale differences. We have now added a paragraph in the Methods to state this explicitly and included a brief reminder in the results. 

Changes to manuscript: Insert in the final paragraph of the Data Reprocessing/Feature Engineering Subsection (2.1) in Methods (after you describe encoding and imputation)... No explicit feature normalization or standardization was applied. After preprocessing, most predictors were binary indicators or bounded clinical scores on comparable numeric ranges, and several primary algorithms used (e.g., tree‑based models) are relatively insensitive to feature scale. We acknowledge that algorithms such as KNN and SVM can be affected by differences in feature magnitude; future work will evaluate whether incorporating scaling materially alters their performance. 

Comment: How were class imbalances handled (if any) during model training and evaluation? Was any reweighting or resampling applied? 

Response: We thank the reviewer for pointing out the issue of class imbalance and the potential use of resampling techniques such as SMOTE or class weighting. While we recognize the value of these methods in certain ML contexts, we chose not to apply them in this study in order to preserve the original class distribution and maintain a clinically realistic evaluation setting. Our goal was to assess how well models would perform without artificial correction for prevalence, especially since high false positive rates are a major concern in early disease screening. We have added a brief rationale for this design choice at the end of Section 2.5. We appreciate the reviewer’s suggestion, which allowed us to clarify an important aspect of our experimental framework.

Changes to Manuscript: At the end of Section 2.5: Prediction and Evaluation Metrics, just before Section 2.6… Although techniques such as Synthetic Minority Over-sampling Technique (SMOTE) or class weighting are commonly employed to address class imbalance, we elected not to apply these approaches in the present study. Our objective was to evaluate model performance under clinically realistic conditions using the raw class distributions present in the dataset. This decision was made to avoid artificially altering the decision boundary or biasing the models in ways that may not reflect actual screening settings, where the prevalence of Parkinson’s Disease is naturally low. Instead, we relied on a three-way data split with stratified sampling to preserve class proportions and used confusion matrix analysis to directly evaluate the consequences of imbalance, particularly in terms of false positive rates. This design choice allowed us to highlight the limitations of current ML approaches in early PD detection without relying on performance enhancements that may not generalize to real-world deployment.

Comment: Have the authors explored why certain models (e.g., DNN) showed slightly better specificity despite overall poorer F1? Could this be due to regularization or architecture design?

Response: We appreciate the reviewer for highlighting the nuanced behavior of the DNN model, which indeed achieved better specificity despite an overall lower F1 score. In revising the manuscript, we expanded this analysis in Section 3.3 to include other models—such as AdaBoost and KNN—that displayed a similar pattern. We now discuss how architectural and algorithmic factors (e.g., dropout and weight decay in DNNs, instance sensitivity in KNN, and iterative weighting in AdaBoost) may have contributed to reduced false positives by promoting more conservative classification thresholds. These insights enrich the discussion of model behavior in subclinical prediction tasks and emphasize the importance of interpreting performance through both confusion matrix analysis and understanding model design. We appreciate the opportunity to clarify and deepen this portion of the analysis.

Changes to Manuscript: Add to the end of section 3.3… Notably, while the deep neural network (DNN) model demonstrated lower overall F1 and accuracy scores compared to simpler models, it achieved the lowest false positive rate among all models evaluated without overt features (Figure 4). This paradoxical pattern—relatively better specificity despite poorer aggregate metrics—was also observed in models such as AdaBoost and KNN, both of which reported modest F1 scores but demonstrated improved ability to correctly identify healthy controls compared to models like LASSO or logistic regression. This phenomenon may be partially explained by differences in model architecture and regularization behavior. For instance, the DNN’s use of dropout and weight decay likely encouraged a more conservative decision boundary, reducing overconfident misclassification. Similarly, KNN’s instance-based learning approach is sensitive to local density, which may yield cautious predictions in sparsely populated control regions. AdaBoost, by emphasizing misclassified instances through iterative weighting, may have inadvertently focused on recovering specificity at the expense of recall. These findings suggest that certain model types, particularly those with non-linear representation capacity or adaptive weighting mechanisms, may be better suited to managing class imbalance in subclinical scenarios where overt signals are absent. However, this increased specificity does not necessarily equate to superior clinical utility, as most of these models still exhibited substantial false positive rates overall.

Comment: It is recommended to discuss recent DL papers like A Hybrid Deep Learning Approach for Bearing Fault Diagnosis Using Continuous Wavelet Transform and Attention-Enhanced Spatiotemporal Feature Extraction and A new dual-input CNN for multimodal fault classification using acoustic emission and vibration signals.

Response: We thank the reviewer for recommending the inclusion of recent deep learning papers on hybrid approaches and multimodal fault classification.

Changes to manuscript: In response, we have expanded our discussion of future directions to highlight how these advances—such as the integration of time-frequency analysis with attention-based deep learning architectures, and the use of dual-input CNNs for combining multiple sensor modalities—offer promising strategies for improving early disease detection. Specifically, we now discuss how lessons from these industrial fault diagnosis studies may inform the development of more robust and sensitive clinical models, particularly as richer physiological data sources become available. 

Comment: In Figure 6, how do the authors interpret the persistent validation gap in KNN even with overt features included—could this suggest a need for additional regularization or feature engineering?

Response: A consistent gap between training and validation accuracy is expected and, to some extent, necessary: the validation curve should be slightly lower than the training curve, as it reflects the model’s ability to generalize to unseen data. In well-behaved learning scenarios, both curves should converge and plateau together as training set size increases, with only a small, stable gap between them. In our results, the curves do converge and stabilize, indicating that the model is not continuing to overfit as more data is added. The observed gap reflects the model’s capacity to fit the training data more closely than the validation data, which is typical for KNN and similar algorithms, particularly when the feature set includes highly predictive variables.

Comment: Given the observed overfitting in high-capacity models, have the authors considered early stopping or dropout (for DNNs) as mitigation techniques?

Response: As shown in Figures 6 and 7 and discussed in Section 3.5 and 4.1, the persistent gap between training and validation accuracy in high-capacity models (e.g., Random Forest, AdaBoost, DNN) is not simply a result of insufficient regularization or premature stopping. Rather, it reflects a fundamental limitation imposed by the feature set: when overtly diagnostic features are included, models can easily memorize the training data, but this does not translate to improved generalization because the task itself becomes artificially easy due to data leakage. Conversely, when these features are excluded, all models—including those with strong regularization—plateau at modest accuracy levels, indicating that the available features lack sufficient predictive signal for early PD detection.

Specifically for the DNN, we already implemented dropout and weight decay, and hyperparameters were tuned using a dedicated validation set (see Methods 2.3). The learning curves show that, even with these measures, the validation performance does not improve with more data or further regularization, but rather plateaus at a level determined by the intrinsic information content of the features. Early stopping would have simply halted training at the same plateau, without closing the gap or improving generalization.

Changes to manuscript: Overall, the revised manuscript already addresses this issue clearly and thoroughly. But we have added (in the Limitations section) the following to further highlight this point for clarity: "It is important to note that additional regularization techniques, such as more aggressive dropout or early stopping, would not have materially altered these findings. The persistent gap between training and validation accuracy in high-capacity models reflects the intrinsic limitations of the available feature set, rather than insufficient regularization or premature stopping. This underscores that the primary barrier to improved generalization is the lack of predictive signal in the features, not the choice of model or training protocol."

Comment: As the dataset comes from a single source, how do the authors plan to extend their findings to more diverse patient populations or multi-center datasets?

Response: We thank the reviewer for highlighting the important issue of generalizability. In response, we have added a discussion to Section 4.3 acknowledging the limitation of using a single-source dataset. We agree that expanding to multi-center or community-based datasets is essential for robust clinical translation. We have outlined plans to pursue external validation and explore advanced strategies such as domain adaptation and federated learning to address potential heterogeneity across patient populations and institutions. We appreciate the opportunity to expand on this critical aspect of future work.

Changes to Manuscript: Add to the end of Section 4.3: Limitations and Weaknessess… A key limitation of the present study is the use of a dataset derived from a single source. While the controlled and standardized nature of this dataset allows for consistent feature collection and preprocessing, it may limit the generalizability of findings across more heterogeneous clinical populations. Differences in demographics, comorbidities, recording equipment, and diagnostic protocols could all influence model performance when applied outside the original context. As a next step, we plan to validate our models on external datasets from multiple clinical sites or community-based cohorts. This would allow for a more comprehensive assessment of generalizability and may reveal population- or site-specific adjustments needed for optimal model performance. Furthermore, future work will explore domain adaptation techniques and federated learning to address inter-site variability while preserving data privacy.

Comment: Do the authors suggest any framework for clinicians to evaluate ML tools for early diagnosis in practice, especially in the absence of technical expertise?

Response: We thank the reviewer for raising this important and practical question regarding how clinicians—particularly those without technical training—can evaluate machine learning tools for early diagnosis.

Changes to Manuscript: We have added a new section {Ensuring Clinical Value} entirely dedicated to communicate this point.

Round 2

Reviewer 1 Report

Comments and Suggestions for Authors

The authors have addressed all the concerns raised by the reviewer. The manuscript may now be accepted.